# Affordance Extraction with an External Knowledge Database for Text-Based Simulated Environments

## Abstract

Text-based simulated environments have proven to be a valid testbed for machine learning approaches. The process of affordance extraction can be used to generate possible actions for interaction within such an environment. In this paper the capabilities and challenges for utilizing external knowledge databases (in particular ConceptNet) in the process of affordance extraction are studied. An algorithm for automated affordance extraction is introduced and evaluated on the Interactive Fiction (IF) platforms TextWorld and Jericho. For this purpose, the collected affordances are translated into text commands for IF agents. To probe the quality of the automated evaluation process, an additional human baseline study is conducted. The paper illustrates that, despite some challenges, external databases can in principle be used for affordance extraction. The paper concludes with recommendations for further modification and improvement of the process.

## 1 Motivation

Simulated environments are an indespensible tool for modern studies of artificial intelligence and machine learning in particular. As such, they provide the possibility to develop new methods and even paradigms, as well as the opportunity for testing and evaluation. Simulated environments come in different modalities, degrees of complexity and fields of applications. Recently, the domain of video games emerged as a popular test field for algorithms, including many concepts, tools and paradigms of modern AI research like Reinforcement Learning (RL), Knowdlege Graphs (KG) and Neural Networks (NN). The potential of these developments has been demonstrated in several landmark studies, for example for the classic board game Go (Silver et al., 2016), or their successful application to Atari games (Mnih et al., 2015). In many of these cases, algorithms were able to achieve or even surpass human performance level.

Encouraged by these results, attempts were made to extend the developed methods to the field of text-based games. Among these, in particular Interactive Fiction (IF) (also known as *text adventures*) received some attention. Environments like *Textworld* (Côté et al., 2018) or *Jericho* (Hausknecht et al., 2019a) were designed specifically to aid studies of machine learning performance on IF games. Correspondingly, challenges have been established to compare the performance of different algorithms in solving selected text based tasks.

However, according to the evaluation metrics of these challenges, even the best performing solutions still rarely achieve scores comparable to human performance without task-specific fine tuning. For example, referring to the first installment of the popular IF franchise *Zork*, up to our knowledge the best results in playing Zork1 (without employing game specific methods) have been achieved by Ammanabrolu & Hausknecht (2020), scoring around 12% of the achievable reward. While certain solutions score impressive results in specified challenges (i.e. *First Textworld Problems* [1]), these solutions are often specialized on particular tasks (i.e. cooking meals) and generalize poorly to other domains (see i.e. Adolphs & Hofmann (2019)).

---

[1] First TextWorld Problems: A Reinforcement and Language Learning Challenge, https://www.microsoft.com/en-us/research/project/textworld/

A major aspect for explaining this performance gap is the increased environmental complexity of IF games compared to other AI challenges, like arcade games or conventional board games. To provide the player with a feeling of immersion, IF simulates an environment which aims to resemble a life-like, detailed scenario for the player to explore and interact with. The emerging setup poses two additional challenges:

On the one hand, all sensory input and communication between agent and environment is mediated by textual descriptions. To convert these descriptions into computationally processable information, methods of natural language processing (NLP) are necessary.

On the other hand, the growing complexity of the environment may inflate both action- and state-space of model descriptions involved, i.e. (partially observable) Markov Decission Processes, (PO)MDPs. For example, collecting all combinatorially possible commands from the parser vocabulatory of *Zork* yields approx. 200 billion commands. Only a small subset of these commands can actually be executed by the parser, while most commands are comprised of arbitrary combinations of verbs, prepositions and objects which do not yield any reasonable semantics. The above mentioned platforms mostly address this problem by introducing some kind of predefined command templates. For a more generalized approach, the concept of *affordances* can be introduced to the field.

In a general sense, *affordances* refer to the possible interactions available to a given agent in any given situation. A seminal work for affordances in the context of IF games is Fulda et al. (2017), and the following arguments build up on the definitions of this work. The corresponding process of *affordance extraction* from external (that is: not game related) knowledge bases is the central focus of this paper.

The following chapters establish and evaluate a way of including external knowledge bases (demonstrated with *ConceptNet*) for affordance extraction in text-based tasks. Three main aspects are addressed:

- The paper aims to demonstrate that IF and to some extent text-based environments in general, can be used as testbeds for automated affordance extraction procedures.

- It is further demonstrated that the process of automated affordance extraction can benefit from external databases like ConceptNet.

- The paper discusses remaining open issues and possible extentions, as well as giving recommendations for future data collection strategies and algorithmic improvements.

## 2 Affordance Extraction

The term *affordance* was introduced by J.J. Gibson in the context of visual perception studies (Gibson, 1966). It was used to describe the situational nature of interactions between an animal and its environment, i.e.: While a monkey is able to climb a tree, an elephant instead can simply reach for its fruits. Both animals can "afford" different actions with the same object. In a more abstract sense, affordances determine interaction options for agents in their environment. Correspondingly, this term has found its way into various fields of research, i.e. robotics, visual computing and natural language processing, as inGrabner et al. (2011); Dubey et al. (2018); ZhjZhu et al. (2014).

For the purpose of this paper, the latter is of central importance. Most algorithms used for solving IF challenges rely on RL and therefore policy generation/optimization. This approach requires a set of possible actions (which can be seen as instances of *affordances* in this context) for every game state. The process of generating such affordances from any provided input (i.e. text excerpts) is called *affordance extraction*.

While affordance extraction recently drew increasing attention from the scientific community (Fulda et al., 2017; Loureiro & Jorge, 2018; Khetarpal et al., 2020), in practical applications this process is sometimes shortened or even omitted and a stronger focus is laid on the rating of available actions. This is possible, because either the available actions are few and trivial (i.e. moving directions in a maze), or they are provided by the environment itself (i.e. *TextWorld* and *Jericho* both provide a set of *Admissible Commands* (AC)). However, for any more general approach using RL and/or (PO)MDP-like models in object-based environments, affordance extraction is a useful and sometimes even necessary technique.

It should be noted that the process of affordance extraction sometimes is not clearly distinguished from policy optimization in literature. That means that by defining an action space, often contextual information, i.e. about the task to be solved, is already included and acts as a filter for affordances. Regarding pure affordance extraction however, the question "Which action could be executed on/with this object?" should not be confused with the question "Which action on this object would make sense to solve the task?". In the scope of this work a clear focus is put on the first question, while the goal to provide a reasonable or even "optimal" action space for solving the game itself is not addressed.

## 3 Interactive Fiction

In this paper, games of *Interactive Fiction* (IF) are used as a test case for affordance extraction. As such, the general mechanics and properties of IF are shortly described in this chapter.

The term *Interactive Fiction* describes a certain genre of video games, in which the player is performing actions in a simulated environment. The actions, as well as the description of the environment, are communicated through a text-based interface. Therefore, this type of games is sometimes colloquially refered to as *text adventures*. The first verified textadventure *Colossal Cave Adventure* was created by Will Crowther in 1975 (Crowther, 1976), with Don Woods expanding and releasing it later in 1977 [2]. Soon after, text adventures grew more popular and a variety of prominent commercial and non-commercial games have been released. Among them was the popular *Zork* franchise, which up to this day provides an often studied challenge for machine learning algorithms (Atkinson et al., 2018; Ammanabrolu et al., 2020; Tessler et al., 2019; Jain et al., 2019; Yin & May, 2019).

While in theory the whole set of natural language can be used to issue commands to an IF game, the parser of the game is limited to certain vocabulary and grammar. Two common types of IF games are distinguished according to the form of command input:

- *Parser-based*: The commands are entered in free text by the player and are then interpreted by a parser.

- *Choice-based*: The player is chosing from a predefined set of commands in every game state. The representation of these commands and their amount may vary (i.e. in form of hyper-text or a plain list of text commands).

In many cases in literature, this distinction is not explicitly made. As for example noted byZelinka (2018), within the limits of most parsers, any parser-based problem can be converted into a choice based one by spelling out explicitly all possible combinations of vocabulary accepted by the parser. This paper will focus exclusively on parser-based IF games.

For the purposes of this paper, IF provides an interesting opportunity to test affordance extraction using external databases. The simulated environment of the games is built around objects and interactions: Along with the movement of the agents (and sometimes dialogues), object interaction is often the only available action and the text descriptions are usually designed with these interactions in mind. This marks an important difference to other text-based media, e.g. belletristic literature, which often only implicitly (or not at all) contain information about a physical environment and interaction possibilities of characters. On the other hand, compared to agents interacting with physical reality, IF games offer a decreased complexity level due to a reduced amount of information about environment and objects, which is mediated by the text.

### 3.1 Environments for Studying Interactive Fiction

Several platforms have been developed to study NLP or machine learning related problems with IF. Among the most prominent are *TextWorld* and *Jericho*, which are also used for the evaluation in this paper.

---

[2]https://www.ifarchive.org/indexes/if-archiveXgamesXsource.html

### 3.1.1 TextWorld

*TextWorld* (Côté et al., 2018) "is an open-source, extensible engine that both generates and simulates text games" [3], developed by Microsoft. It was specifically designed as a testbed for reinforcement learning agents in the context of language understanding and (sequential) decision making. Therefore, it provides important features valuable for studies of IF:

- Customizable scenarios: By adjusting input parameters, the user is able to create scenarios with varying degree of complexity, i.e. the number of involved objects or the minimal amount of steps to solve the scenario.

- Arbitrary large test/evaluation set: In principle, a very large amount of scenarios can be generated by TextWorld, limited only by the number of items, objects and locations in the vocabulary of the engine.

- Different game modes: By offering three different goal definitions (Cooking a meal, collecting coins and finding a treasure), TextWorld theoretically allows for the evaluation of different skillsets and a very rudimentary kind of generalization.

- Observation- and evaluation options: The TextWorld engine contains a set of options to assist studies of agents interacting with the scenarios. For example, lists of available objects and commands for any situation can be obtained, as well as information about location, score and even an optimal "walkthrough" towards the game's goal.

Building on these features, TextWorld has been featured in a number of studies (i.e. Jain et al. (2019); Hausknecht et al. (2019b); Tao et al. (2018); Zelinka et al. (2020); Madotto et al. (2020)) and, most notably, it has also been used to facilitate the *First TextWorld Problems* competition (*FTWP*, see Chapter 1).

### 3.1.2 Jericho

*Jericho* (Hausknecht et al., 2019a) is another platform for language and AI based studies of IF, but it follows a rather different approach than TextWorld. Instead of creating its own IF scenarios from basic pieces of vocabulary and scenario related input parameters, the Jericho suite is compiled from already existing IF and is made available through the Jericho framework. The platform supports over 30 textadventures, which cover a wide range of commercial and non-commercial games (i.e. fan projects), spanning several decades. Similar to TextWorld, Jericho provides options to access information about the full game state (including inventory, score and ACs) as well as walkthroughs for supported games.

While in principle both frameworks offer similar options for controlling an agent and evaluating corresponding moves, there are important differences regarding the nature of the scenarios itself. While TextWorld focusses on simplified and fully customizable sets of scenarios for the specific purpose of RL/ML studies, the IF games in Jericho were created for the purpose of human entertainment. As such, the latter tend to have a higher degree of complexity and creativity as they are meant to challenge and entertain a human user. On the contrary, the solution to TextWorld scenarios is often perceived to be very easy or even obvious by human players. In this respect, the somehow complementary nature of both platforms offers an opportunity to study the performance of algorithms for different levels of complexity.

## 4 Simulation Setup

### 4.1 External Knowledge and ConceptNet

In the context of RL related approaches, data is usually collected through interactions of an agent with its environment. To generate data, information on available interactions usually has to be supplemented and

---

[3]https://www.microsoft.com/en-us/research/project/textworld/

can rarely be created by the algorithm itself. Often, the definition of available actions is made explicitly, especially if the action space is small and consists of only a few actions (see Section 2). For more complex environments, like IF related games, this approach becomes more difficult. The algorithm has to either rely on pre-programmed generalisation rules (i.e. "every object can be taken"), pre-defined sets of commands, or explicit knowledge about every available object. The latter can rarely be supplied completely by human authors for every scenario. In this case, utilizing an existing knowledge database about objects and their properties is able to fill gaps and provide additional affordances.

Surveying databases as a possible resource for affordance extraction, several criteria have to be fulfilled:

- Data density: The resource should contain a large amount of information about objects and their usage.

- Data diversity: The resource should not focus on a specific context or scenario to ensure a certain generalizability.

- Data accessability: The information should be available in a machine readable format or at least be convertible without significant information loss.

- Efficiency: The data extraction process should have a suitable time performance for testing on a large number of scenarios.

- Data availability: The data from the resource should be accessible during the whole run time, either via web API or as a (feasible) local copy.

Several data bases have been investigated to be used for affordance extraction:.

- WikiData (Vrandečić & Krötzsch, 2014) is an extension of the online encyclopedia Wikipedia. It adds a machine readable knowledge graph to a lot of wikipedia's topics. It also contains information about the typical usage of objects, denoted using the *use* property. WikiData can be queried through a publicly available API.

- WordNet (Fellbaum, 1998) organizes terms from the english language in a semantic network. Word-Net supports disambiguation of terms, by using *Synsets* (sets of synonyms, forming a common concept) and explicitly orders terms along other relations, such as *antonymy, hyponomy, meronymy, troponymy* and *entailment.* None of these relations refers to the typical usage of an object or a term.

- ConceptNet (Speer et al., 2016) is a partially curated collection of commonsense knowledge from a number of sources. Among other sources like DbPedia, OpenCyc and others, the Wiktionary online dictionary is used to gather its contents. ConceptNet organizes its contents in a knowledge graph using a set of relations[4] to connect nodes within the graph. ConceptNet does not offer disambiguation of terms. Among the supported relations (see Speer et al. (2016) for more details) some explictly address the common usage of objects, such as: *CapableOf, UsedFor, ReceivesAction.* The knowledge graph of ConceptNet is accessible via a public available API.

- NELL (Mitchell et al., 2018) is an acronym for "Never Ending Language Learner". It refers to a continually updated knowledge base of terms, which are ordered in a taxonomy. Updates of the knowledge base happen mostly automatically by means of web-crawlers parsing sites on the internet and storing the parsed information as logical statements within the taxonomy. During preliminary tests, no useful information for affordance extraction could be extracted from this knowledge base.

Referring to the above mentioned criteria, *ConceptNet* (CN) has been selected for evaluation.

---

[4]https://github.com/commonsense/conceptnet5/wiki/Relations

For the purpose of this study, CN has some major advantages: It provides information in a machine readable format, as well as an API for automated queries. At the time of this study CN provides about 34 million pieces of relational information (*edges*) and therefore offers a comparatively large amount of possible environmental information. The next Section discusses the CN properties relevant for this study.

## 4.2 Command Generation with ConceptNet

The object information in ConceptNet is organized into labeled categories like synonyms, antonyms or typical locations, where the object can be found. For the extraction of possible object affordances, in principle three categories are offering corresponding information:

- *used for*: Describes what the object is *used for*, usually resulting in a single verb phrase (e.g. "knife is used for slicing").

- *capable of*: Describes what the object is *capable of*. Semantically, this often yields results similar to the *used for*-category, but puts more emphasis on active capabilities, rather than the passive usage.

- *receives action*: Describes which action can be performed *on* the object. In the description of ConceptNet, this category is somehow misleadingly labeled "Can be..." (see Section 6.2). An example for this category is "a book can be placed in a drawer".

A preliminary analysis showed that most relevant information for affordance extraction is concentrated in the *used for*- and the *receives action*- categories. The *capable of*- category often contains information which is either irrelevant or already covered in the other categories. Furthermore, ConceptNet offers weights (reflecting the frequency of appearance) for every entry. While weights are not used in this proof-of-concept study, they are in principle a useful option, as the human input sometimes refers to subjective or (very) rare affordances (for example, one user provided the entry "knife is capable of free one's soul"). These special cases may provide useful for some studies which try to deal with "long tail events" or advanced reasoning, but for the purpose of affordance extraction on a basic level they are ineffective most of the time.

The information stored in ConceptNet is organized in n-tuples of information, connecting concepts via the previously discussed categories above. These n-tuples (mostly triples, i.e. "(knife, UsedFor, cutting)") in their unaltered form cannot be interpreted by any IF parser as commands for the agent. Some means of translation have to be established. This task itself is related to Natural Language Generation (NLG) and many sophisticated solutions might be employed in future applications. For the purpose of this paper, some basic translation rules are defined. In the affordance extraction algorithm, two rules are used, referring to the number of items involved. Conveniently, these two rules correspond to the two categories selected for evaluation in this section:

### 4.2.1 Affordances with one object

Information from the *Receives Action* category usually refers to actions which can be performed on the queried object. The corresponding template for triples of this category is therefore designed as: verb (imperative form) + object, i.e. the triple "(door, ReceivesAction, opened)" transforms to "open door".

### 4.2.2 Affordances with two objects

The information from the *Used for* category is phrased in a more modal way. For example the triple "(knife, UsedFor, slicing)" implicitly refers to another object, which can be sliced with a knife. By identifing the required second object, an affordance with two objects is formed, i.e. "(knife, UsedFor, slicing, tomato)". When querying ConceptNet, the returned text fragments are processed with spaCy (Honnibal & Montani, 2017) to determine if, in addition to the object in question and a verb, more nouns are present. If this is the case, these nouns are considered to be additional objects, that are required for the described action. If two objects of such an affordance are available at the same time, a

corresponding command becomes possible. The translation rule for this case is defined as: verb (imperative mood) + target object + preposition + queried object, i.e. the 4-tuple "(knife, UsedFor, slicing, tomato)" transforms to "slice tomato with knife". The correct proposition is determined via statistical means by counting associated prepositions for every word in large text corpora. While this solution is sufficient for the study at hand, it is modularized and simply exchangeable with a more sophisticated routine.

It should be mentioned, that the command generation is largely dominated by one-object-affordances, due to the comparativly sparse connections between most objects in ConceptNet. Therefore, two-object-commands are rarely relevant for later evaluation. Abstraction techniques might address this problem (See Section 6.2).

### 4.3 Algorithm Overview

The algorithm is written in Python 3.6. Its general strucuture for the automated evaluation procedure is shown in Figure 1.

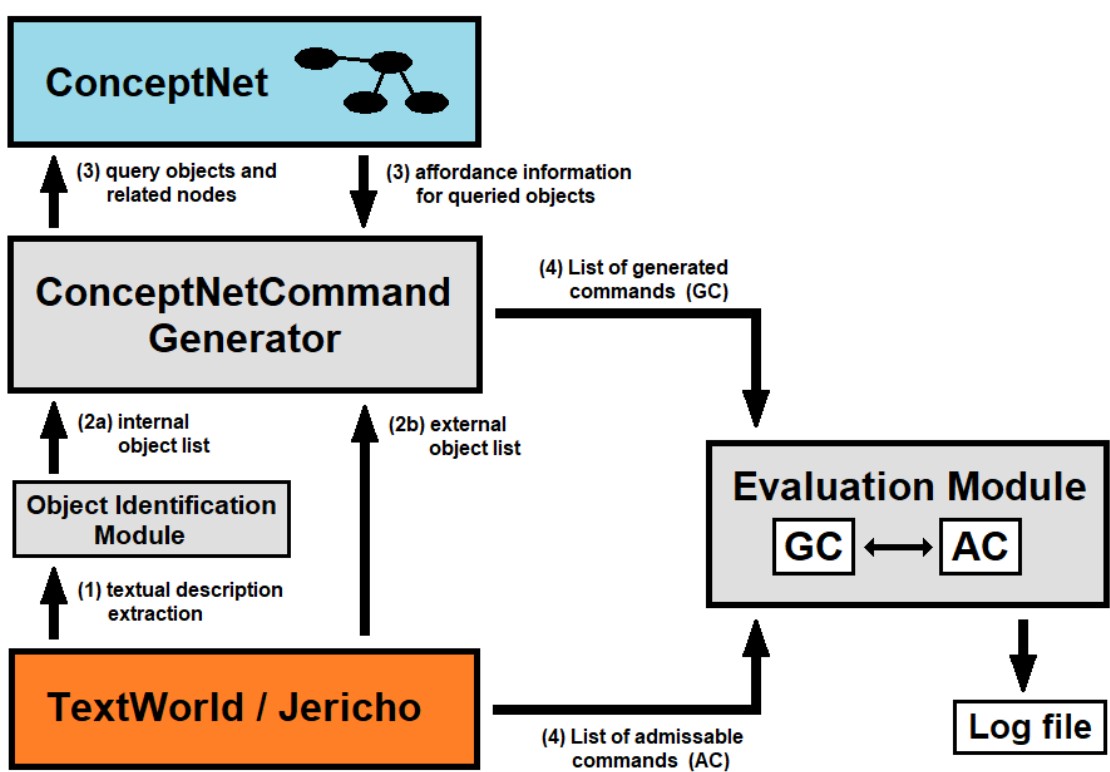

Figure 1: Algorithmic setup for the automated evaluation of the generated commands.

The algorithm is designed to recieve a piece of text as input and is therefore able to be used with any text source. For the purpose of this study an interface to TextWorld and Jericho (see Section 3.1) is assumed. The algorithm performs the following steps:

(1) **Textual Description Extraction**: Entering a new scenario, the algorithm receives textual information about the environment using the "look" command (or parsing the general description provided by the

game), as well as textual information about the available inventory. Both inputs are concatenated into one string. Note, that for this step in principle every text source is valid. Textual description extraction then has to be adjusted accordingly.

(2) **Object Identification**: From the textual input of (1), objects for possible interaction are identified. In some cases, it is possible to use pre-defined lists of objects for this task (this is done when playing in TextWorld, denoted as (2b) in Figure 1). But as the algorithm is designed to work for every text description, identification of objects via PoS-Tagging is also available as a more general method (this is done when playing in Jericho, denoted as (2a) in Figure 1). The algorithm uses spaCy as a *Parts of Speech-Tagger* (PoS-Tagger). It should be noted that this solution is not particularly robust against certain situational and linguistic problems (i.e. recognizing a boat on a painting as an actual item in the room or that "open window" can be interpreted as a window that is open, or as a command to open a closed window). For the purpose of this paper, these disadvantages are accepted to retain a broader field of application.

(3) **Object Affordance Query**: The list of objects is handed to the command generator module (named *ConceptNetCommandGenerator*), which queries corresponding information from ConceptNet. The criteria of ConceptNet regarding queried categories and weights are described in Section 4.2. While querying ConceptNet, a local cache is used to speed up future queries in the evaluations of this study. This cache is however entirely optional. Alternatively the query cache can also be substituted with a full knowledge graph, using for example a RDF data format.

(4) **Command Generation**: The command generator module converts the queried affordances into text commands according to the translation rules stated in Section 4.2. As a result, a list of generated commands (GCs) is handed to the evaluation code, along with other relevant information from the corresponding state (i.e. ACs provided by the framework).

(5) **(Automated) Evaluation**: The GCs are evaluated automatically according to the procedure and metrics outlined in Sections 5.1 and 5.2. The results are stored into a logfile. For the human evaluation procedure (see Section 5.4), this step is slightly modified. The ACs are discarded, and the GCs from step (4) are instead presented to a human volunteer in pair with the corresponding textual description from (1).

(6) **Proceeding to the next Scenario**: The algorithm executes a command selected from a predefined list (usually a walkthrough) to enter the next scenario. Step (1) commences, until all pre-selected scenarios of the game have been visited. This step is not visualized in Figure 1 and its execution depends on the applied text source.

The algorithm is modularized and allows for improvement or replacement of single components, i.e. object identification (Step (2a)) or command generation (Step (4)).

## 5 Evaluation

### 5.1 Evaluation setup

To assess the performance and the robustness of the algorithm in an efficient way, the evaluation setup aims to present as many unique and clearly distinguishable scenarios as possible to the agent. Naturally, different games (usually) offer different scenarios, so for both frameworks, Textworld and Jericho, a sufficient number of games is selected to maintain diversity (see Section 5.3). However, it should also be ensured that the agent does not evaluate too many similar scenarios within a single game, i.e. by getting stuck in the same room. The walkthrough option provided by both frameworks offers a simple way to automatically achieve progression in the current game setup. The walkthrough is provided as a list of commands, which guide the agent towards the corresponding goal when executed sequentially. While following the walkthrough path, the

steps for affordance extraction and command generation outlined in Section 4.3 are executed. Additionally, the extraction and generation process is only applied, if the location of the current state has not been visited before, so location induced double counting (for example when travelling between locations and passing the same site multiple times) is eliminated.

## 5.2 Evaluation metrics

The routine described in Sections 4.3 and 5.1 produces a list of affordances (phrased as parser-compatible text commands) for any textual input. To assess the quality of these results, two evaluation types are employed: The automated comparison to the ACs provided by the framework, and a human baseline.

The automated comparison refers to the option available in both TextWorld and Jericho to provide a list of commands which can be executed in the current game state. The traditional metric of *precision* can be applied to this situation: This metric quantifies, how many of the produced commands are also an AC. If the affordance extraction process reproduces exactly the AC list, a score of 1 is achieved.

However, this kind of metric alone is not sufficient to judge the quality of the generated affordances. The reason for this lies in parser related constraints (this refers to grammatical patterns as well as vocabulary) and game related simplifications. The AC list is by no means complete. In every situation, there are additional commands that would appear reasonable by human standards, but will not be recognized by the parser and thus are not listed as an AC. This can refer to commands which are not part of the parser's vocabulary due to the use of synonyms (e.g. "shut door" instead of "close door") as well as actions which have not been taken into account when the game was made, mostly because they are not relevant for the task-related progress within a game (i.e. "carve table with knife" or "jump on table"). For this reason, a human baseline is introduced in addition to the automatically computed metrics: A volunteer ([one of the authors, anonymized for review process]) is presented with the textual description of the scenario, as well as the list of commands generated from this description. The volunteer then is asked to sort each command into one of the following categories:

- Contextually suitable (A): The command is perceived as physically possible and useful in the current situation.

- Valid, but contextually infeasable (B): The command is generally perceived as valid, but does not appear useful or feasible in the current situation (e.g. "bite table"; or "open window" if the window is not physically present, but only mentioned in a dialogue).

- Invalid (C): The command is physically impossible or does not make sense grammatically (Examples: "put house in your backpack", "drink book", "open door from key").

For the interpretation of the results the subjective nature of the answers needs to be taken into account. Yet, the human baseline offers a broader perspective on the performance of the algorithm which is not limited by the NLP related capabilities of the parser and the game design.

## 5.3 Evaluation on Jericho and Textworld

The algorithm is first evaluated on the Jericho platform. A total of 37 games have been selected for evaluation, containing 1226 evaluation scenarios overall. The first line of Table 1 shows the amount of generated commands and matches with an AC command, as well as the corresponding precision percentage. A more detailed compilation of results for every single game is given in Table 5. In 24 of 37 games, at least one AC command is reproduced with results varying in an intervall between 0 and a maximum of 9 commands (1.83% precision, achieved in the game "hollywood"). In most cases, only a small absolute amount of matching commands is produced, with a strong focus on recurring commands like "open/close door". Overall, with only 0.4% of the generated commands matching an AC, the unmodified algorithm is not yet able to produce

a significant set of affordances to be used with the game parser. The reasons for this behavior and possible improvements are studied later in this section, with the current results serving as a baseline.

Table 1: Evaluation results for the basic algorithm (first line) and the "take"-addition on the Jericho testset (second line). The columns denote the steps, the generated commands, the total amount of generated commands matching the ACs and the corresponding percentage of matching commands for every game.

| Evaluation mode | Steps | Gen. Com. | Corr. Com. | Corr. Com. (%) |
|---|---|---|---|---|
| Jericho base | 1226 | 12949 | 52 | 0.4 |
| Jericho add take | 1226 | 17743 | 149 | 0.84 |

The next evaluation step addresses the problem of trivial background knowledge. The affordances provided by ConceptNet tend to focus on rather complicated and creative activities, while mostly omitting "trivial" ones (see Section 6.2). The second line of Table 1 shows another evaluation on the same testset, with a simple "take" affordance added to every identified object (implying that most objects can be picked up). The complete results for every single game are given in Table 6. Now, only for 3 out of 37 games no AC is reproduced. Overall the amount of correctly reproduced ACs nearly increases by a factor of three, from previously 52 matching commands up to 149 (0.84%). The distribution among single games remains uneven, reflecting the heterogeneous character of the test games: While for few games still no matching command is produced at all, up to 17 matching commands for "hollywood" or 13 commands for "zork1" and "zork2" are generated. It should be noted that the overall precision does only improve by a factor of two, as by indiscriminately adding a "take" affordance to every object, additional noise is produced.

The next evaluation step addresses the heterogeneity and the entertainment focus of games included in the Jericho platform. By applying the algorithm to a set of structurally simpler TextWorld games, the influence of this kind of complexity on the result can be assessed. For this evaluation run, 333 TextWorld games with a total of 984 evaluation scenarios have been created. The results for the evaluation with the standard algorithm (first line), as well as the "take"-addition" (second line), are depicted in Table 2.

Table 2: Evaluation results for the basic algorithm (first line) and the "take"-addition (second line) on the TextWorld testset. The columns show the steps, the generated commands, the total amount of generated commands matching the ACs and the corresponding percentage of matching commands for every game.

| Evaluation mode | Steps | Gen. Com. | Corr. Com. | Corr. Com. (%) |
|---|---|---|---|---|
| TextWorld base | 984 | 5143 | 330 | 6.42 |
| Textworld add take | 984 | 7726 | 438 | 5.67 |

While providing a roughly similar amount of evaluation scenarios (984 against 1226 for Jericho), the algorithm provides only 5143 generated commands (vs. 12949 for Jericho), mirroring the decreasing diversity and level of detail in the TextWorld descriptions. Of the generated commands, 330 (ca. 6.42% precision) matched an AC, fluctuating between 0 and 3 correct commands for every step. It should be noted, that these results are vastly dominated by "eat" and "close/open" commands and that often correct commands are repeated over several steps (i.e. by carrying an corresponding item in the inventory). Still, the overall percentage of correct commands improves by a factor of around 16. The addition of a "take"-affordance to every item further increases the number of produced commands to 7726, with 438 being recognized as an AC (ca. 5.7% precision). In this specific case, the increase of correct commands is significantly less compared to the Jericho evaluation and the corresponding precision even decreases. This is caused by a special characteristic of the chosen TextWorld testset: The agent starts with a large number of items already in its inventory, rendering them effectively immune to a "take" command.

### 5.4 Evaluation of human baseline

The last evaluation step addresses platform related limitations. To determine whether a generated command could be regarded as "correct", the platform provided ACs have been used. However, these ACs only cover a small set of all affordances available from the game descriptions. Generally the ACs are limited by vocabulary, templates and generation routines. To explore the magnitude of this effect, a selection of three Jericho games ("awaken", "ballyhoo" and "Zork: the Undiscovered Underground" (ztuu)) is used as the input for the affordance generator. The produced commands are then evaluated by a human volunteer according to the categories established in Section 5.2. The corresponding results are depicted in Table 3.

Table 3: Human baseline for three selected Jericho games. The generated commands are sorted into the categorys "Contextually suitable" (A), "Valid, but contextually infeasible" (B) and "Invalid" (C) (See Section 5.2).

| Jericho Game | A | B | C | gen. commands |
|:---:|:---:|:---:|:---:|:---:|
| awaken | 44 | 119 | 113 | 276 |
| ballyhoo | 48 | 136 | 73 | 257 |
| ztuu | 36 | 143 | 69 | 248 |

The results reveal that between 59% ("awaken") and 72% ("Zork: the Undiscovered Underground") of the commands are recognized as an functional affordance by human judgement (referring to categories A or B from Table 3). While being prone to small subjective fluctuations, this marks an significant increase of identified affordances/commands compared to the automated parser evaluation. This has two reasons: Firstly, some commands are simply too exotic for the parser and its vocabulary. Secondly, some affordances were produced by text descriptions not related to actually available items in the scenario (i.e. pieces of dialogue or memories). The last point is quantitatively addressed by distinction between the categories A and B. For all three games, the category B ("Valid, but contextually infeasable") dominates with a factor between three and four over category A, revealing a large influence of "noisy" items not physically relevant/present in the current situation. An illustrative example is given in Appendix B.

In a next step, the evaluation is repeated for the TextWorld setup. To keep human evaluation feasible, only a small subset of 10 scenarios, each from a different game, has been chosen randomly for evaluation (see Table 4). As a result, 69% of the produced commands were deemed either category A or B, mirroring the results from the Jericho evaluation. Between the two functional categories, B still dominates over A, but only with a factor of about two. This again emphasizes the effect of "noisy" language, which in TextWorld is reduced by using the predefined list of interactable objects.

## 6 Summary and Scope

The evaluation in the previous Section showed that external databases can be used to generate affordance for text based input with comparatively little effort. The results however have to be interpreted carefully to account for all upsides and limitations of this simple approach. For the automated evaluation (see Tables 1 and 2) three major observations have been made:

- The total amount and percentage of correctly reproduced ACs for the basic approach is rather low, yielding 52 commands (0.4%) for Jericho and 330 commands (6.4%) for TextWorld. This serves as an illustration point to the many potential challenges of automated affordance extraction.

- The amount of correctly reproduced ACs is increased by a factor of 2.9 (for Jericho) and 1.3 (for TextWorld), respectively, by manually adding trivial "take"-affordances to every evaluation step. This illustrates that information retrieved by external databases might often omit "trivial" information.

Table 4: Human baseline for ten selected TextWorld games. The evaluation only refers to the first scenario of each game. The generated commands are sorted into the categorys "Contextually suitable" (A), "Valid, but contextually infeasible" (B) and "Invalid" (C) (See Section 5.2).

| TextWorld Game | A | B | C | gen. commands |
|---|---|---|---|---|
| Game 329 | 2 | 6 | 3 | 11 |
| Game 108 | 0 | 2 | 3 | 5 |
| Game 140 | 2 | 3 | 3 | 8 |
| Game 32 | 2 | 5 | 0 | 7 |
| Game 62 | 0 | 3 | 0 | 3 |
| Game 157 | 1 | 2 | 0 | 3 |
| Game 155 | 4 | 4 | 4 | 12 |
| Game 55 | 1 | 1 | 3 | 5 |
| Game 160 | 2 | 2 | 0 | 4 |
| Game 171 | 1 | 0 | 3 | 4 |
| Overall | 15 | 28 | 19 | 62 |

- The comparison between the results of Jericho and TextWorld showed a significant increase of the percentage of correctly reproduced ACs (by a factor of ca. 16) for the TextWorld evaluation. This emphasizes the major influence of text complexity and object identification to the result.

Finally, a human evaluation illustrated the limitations introduced by the ACs themselves (see Tables 3 and 4). Compared to the only 0.4% (Jericho) and 6.4% (TextWorld) of affordances being marked as suitable by the automated evaluation, human interpretation deemed between 59% and 72% of all produced commands in Jericho as functional (69% for Textworld), meaning they are either contextually suitable (category A) or at least valid in general (category B). Still the comparative low fraction of contextually suitable affordances hinders practical application at this point. The next subsections address open issues and possible measures for improvement.

## 6.1 Algorithmic aspects

Several components of the algorithm outlined in Section 4.3 feature NLP related processes, which were kept very simple and therefore offer room for improvement. The following points could improve affordance extraction performance by *modifying the algorithm*:

- **Object identification**: The significant difference in performance between evaluations on TextWorld and Jericho testsets (see Section 5.3) highlights the well expected importance of object identification. While TextWorld offered a predefined list of interactable objects, in the evaluation for the Jericho platform objects are extracted via simple PoS-tagging. The algorithm only performs rudimentary post processing. Objects are not contextualized, which means that for example an object in a picture or mentioned in a dialogue will be treated as an interactable object. The conducted evaluation steps already showed the large improvement potential of this measure. Possible solutions might be the implementation of advanced semantic text contextualization techniques.

- **Object ambiguity**: Currently the post processing does not retain adjectives or other describing text, because this would complicate the query process (as ConceptNet only features objects without adjectives). This means that a red and a green paprika will be treated as the same item. Although both of this issues have rather little impact on the study at hand, they should be addressed for more general applications. Possible solutions might be an extended object buffer, maintaining the object properties while dealing with external databases.

- **Forced command patterns**: To convert the affordances extracted from ConceptNet into parsable text commands, predefined translation rules are used. This puts constraints on the grammati-

cal variations of commands. A solution for this problem could be the usage of more advanced Natural Language Generation (NLG) techniques, which can put verbs and objects into reasonable sentence/command structures.

- **Selection of ConceptNet parameters**: While preliminary testing showed that most usable information was contained in the categories *ReceivesAction* and *UsedFor*, information stored in other categories (such as *CapableOf*) is currently not used at all. By including these categories and finding coherent ways to convert this information into text commands, better results are achievable. This also holds for the usage of weights and other porperties offered by ConceptNet (i.e. synonyms).

- **Refined data structures**: Currently, the algorithm directly converts ConceptNet queries to textcommands. The algorithm also provides the option to store obtained information in a knowledge graph using the RDF format. Right now, the only types of edges in this graph are "affords", which connects objects and verbs, and "requires", which is used to denote that an additional object is required for a given affordance (i.e. if one wants to cut something, a tool for cutting and object that is to be cut are required). Given the wealth of information available in ConceptNet and other knowledge bases, it might be useful to use additional means of structuring the local RDF based knowledge graph for affordances. It would i.e. be possible to use type-/instance-relations and subclass-/superclass-relations between nodes of the graph to improve generalization of learned affordances. This path was not followed in the evaluation, as it would have required large additional considerations beyond the scope of a first explorational study. Yet, advanced data storing/processing structures hold large potential for performance improvement, as many knowledge graph related studies mentioned in the introduction show.

## 6.2 Data quality

While ConceptNet is a semantic network designed to boost machine understanding, some aspects of it (and most other similar knowledge bases) still offer potential for improvement with regard to the process of affordance extraction. The following points could enhance affordance extraction performance by improving *data quality/quantity*:

- **Data density**: Despite ConceptNet offering 34 million pieces of relational information (Speer et al., 2016), the frameworks of Jericho and TextWorld contain a multitude of usual and unusual objects. As a result, even in this study many queried objects did not have an entry in CN. The direct connection of two (or more) objects is even more sparse. Therefore, querying rarely offers a combined action of two objects which are found in the IF environment. Further collection of data would correspondingly enhance the performance.

- **Obligation for creativity**: Many of the entries in ConceptNet entered by human operators showed some attempt for creativity. While the most trivial performable actions were not mentioned (probably because they seemed too obvious for the operator), more complicated actions were offered. For example actions like "divide pills" or even "free one's soul" were associated with the item "knife", but the simple facts that a knife can be "picked up" or "put down" were not mentioned. As a solution, such simple actions currently have to be provided to the algorithm by a software engineer. It is advised to add them to the semantic network or the local knowledge graph in the future, either by adapted instructions for human operators, by explicit addition or, to not inflate the amount of information, by employing categorizational rulesets.

- **Unclear instructions**: At several points, faulty information is provided by the network, because the instructions for the human operators are misleading. For example, the category *ReceivesAction* stores information about actions which can be performed *on* a specific item. But to acquire the corresponding data, some human operators were required to complete the sentence "[Item] can be...". It is clear, that the semantic network expects a verbal phrase, but some human operators misread this instruction as a request for properties. For example a data entry for "knife" in the *ReceivesAction* category lists "sharp" as an answer, while rather a verbal phrases like "sharpened",

"thrown" etc. were expected. To avoid this problem, clear and unambigous instructions should be provided for human input.

To conclude, the algorithm is able to produce a fair amount of affordances from textual input by the standard of human judgement. The results are achieved by relatively simple means, using relational input from ConceptNet and some basic generation rules. They can be understood as a proof-of-concept and the listed aspects and arguments can serve as a recommendation and encouragement for further data collection and algorithmic improvement.

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

# A    Detailed Result tables

This section provides more detailed results for automated evaluation.

Table 5: Evaluation results for the basic algorithm on the Jericho testset. The columns show the steps, the generated commands, the total amount of generated commands matching the ACs and the corresponding percentage of matching commands for every game.

| Game | Steps | Gen. Com. | Corr. Com. | Corr. Com. (%) |
|---|---|---|---|---|
| 905 | 6 | 126 | 2 | 1.59 |
| adventureland | 19 | 71 | 0 | 0 |
| awaken | 15 | 276 | 0 | 0 |
| balances | 12 | 210 | 0 | 0 |
| ballyhoo | 38 | 257 | 2 | 0.78 |
| cutthroat | 34 | 396 | 4 | 1.01 |
| detective | 20 | 442 | 0 | 0 |
| enchanter | 48 | 742 | 1 | 0.13 |
| enter | 18 | 320 | 1 | 0.31 |
| gold | 22 | 228 | 0 | 0 |
| hhgg | 29 | 438 | 2 | 0.46 |
| hollywood | 52 | 491 | 9 | 1.83 |
| huntdark | 7 | 80 | 0 | 0 |
| infidel | 46 | 476 | 0 | 0 |
| inhumane | 41 | 338 | 1 | 0.3 |
| jewel | 37 | 379 | 0 | 0 |
| karn | 28 | 259 | 1 | 0.39 |
| library | 7 | 124 | 0 | 0 |
| ludicorp | 79 | 312 | 2 | 0.64 |
| lurking | 48 | 1134 | 2 | 0.18 |
| moonlit | 6 | 126 | 0 | 0 |
| murdac | 74 | 232 | 0 | 0 |
| night | 11 | 58 | 1 | 1.72 |
| omniquest | 31 | 188 | 2 | 1.06 |
| pentari | 16 | 83 | 0 | 0 |
| planetfall | 69 | 531 | 2 | 0.38 |
| plundered | 42 | 471 | 0 | 0 |
| reverb | 17 | 68 | 1 | 1.47 |
| seestalker | 20 | 157 | 1 | 0.64 |
| sorcerer | 63 | 722 | 2 | 0.28 |
| temple | 19 | 431 | 1 | 0.23 |
| wishbringer | 43 | 549 | 3 | 0.55 |
| zenon | 13 | 103 | 1 | 0.97 |
| zork1 | 72 | 658 | 2 | 0.30 |
| zork2 | 62 | 802 | 6 | 0.75 |
| zork3 | 46 | 423 | 2 | 0.47 |
| ztuu | 16 | 248 | 1 | 0.4 |
| Overall | 1226 | 12949 | 52 | 0.4 |

Table 6: Evaluation results for the "take"-addition algorithm on the Jericho testset. The columns show the steps, the generated commands, the total amount of generated commands matching the ACs and the corresponding percentage of matching commands for every game.

| Game | Steps | Gen. Com. | Corr. Com. | Corr. Com. (%) |
|---|---|---|---|---|
| 905 | 6 | 167 | 2 | 1.2 |
| adventureland | 19 | 112 | 2 | 1.79 |
| awaken | 15 | 359 | 1 | 0.28 |
| balances | 12 | 256 | 2 | 0.78 |
| ballyhoo | 38 | 377 | 5 | 1.33 |
| cutthroat | 34 | 571 | 7 | 1.23 |
| detective | 20 | 512 | 3 | 0.59 |
| enchanter | 48 | 1001 | 6 | 0.6 |
| enter | 18 | 442 | 2 | 0.45 |
| gold | 22 | 327 | 5 | 1.53 |
| hhgg | 29 | 554 | 6 | 1.08 |
| hollywood | 52 | 932 | 17 | 1.82 |
| huntdark | 7 | 102 | 0 | 0 |
| infidel | 46 | 680 | 7 | 1.03 |
| inhumane | 41 | 459 | 3 | 0.65 |
| jewel | 37 | 528 | 0 | 0 |
| karn | 28 | 360 | 2 | 0.56 |
| library | 7 | 157 | 0 | 0 |
| ludicorp | 79 | 433 | 2 | 0.46 |
| lurking | 48 | 1464 | 3 | 0.2 |
| moonlit | 6 | 156 | 2 | 1.28 |
| murdac | 74 | 317 | 4 | 1.26 |
| night | 11 | 82 | 1 | 1.22 |
| omniquest | 31 | 286 | 5 | 1.75 |
| pentari | 16 | 20 | 1 | 5 |
| planetfall | 69 | 744 | 5 | 0.67 |
| plundered | 42 | 676 | 4 | 0.59 |
| reverb | 17 | 109 | 2 | 1.83 |
| seestalker | 20 | 236 | 2 | 0.85 |
| sorcerer | 63 | 919 | 5 | 0.54 |
| temple | 19 | 551 | 3 | 0.54 |
| wishbringer | 43 | 725 | 6 | 0.83 |
| zenon | 13 | 140 | 1 | 0.71 |
| zork1 | 72 | 958 | 13 | 1.36 |
| zork2 | 62 | 1104 | 13 | 1.18 |
| zork3 | 46 | 575 | 3 | 0.52 |
| ztuu | 16 | 352 | 4 | 1.14 |
| Overall | 1226 | 17743 | 149 | 0.84 |

## B   Example for human baseline affordance rating

The following example is a game step from "Zork: the Undiscovered Underground").

**Situation description by game**:

Backstage: Ah ah choo. Those curtains. If I weren t so busy helping you with this game, I d suggest you go on without me and let me clean this place up enough so that when you returned, I could at least describe

it decently. I ll do the best I can though. A thick maroon curtain separates the backstage area from the stage. This area was obviously the target of a small underground tornado, a Vorx as scrims scenery and costumes litter the floor. Even an old steamer trunk, virtually decaying from age, rests in a corner. Your inventory: brass lantern, glasses, ZM$100000, Multi-Implementeers, Forever Gores, Baby Rune, razor-like gloves, cheaply-made sword

**Candidate commands generated by the algorithm**:

'cover floor', 'fill glasses', 'find floor', 'find gloves', 'find trunk', 'lie floor', 'live area', 'need glasses', 'play game', 'use lantern', 'wear glasses' (11)

**Contextually suitable (A)**: 'use lantern', 'wear glasses' (2)

**Valid, but contextually infeasable (B)**: 'cover floor', 'fill glasses', 'find floor', 'find gloves', 'find trunk', 'play game' (6)

**Invalid (C)**: 'lie floor', 'live area', 'need glasses' (3)

