# OpenReview forum: "Affordance Extraction with an External Knowledge Database for Text-Based Simulated Environments"
_TMLR — Withdrawn by Authors_

### Review · Reviewer_t4td · 2022-10-19

**Summary Of Contributions:**

This paper explores the use of external knowledge bases as a source for identifying plausible actions in text-based games. The paper proposes a system that (1) identifies mentioned objects in a scene, and (2) uses ConceptNet to extract possible commands in the scene. They evaluate extracted commands automatically by comparing with a set of "admissable commands" (provided by the game), and through human evaluation.

**Broader Impact Concerns:**

I cannot imagine significant negative or positive broader impacts of this work.

**Requested Changes:**

- Evaluation of end-task performance with the proposed method for training text-game RL agents.
- Use of recall as an automatic, instead of / in addition to precision, wrt. the ACs.
- Significantly more content and work in the realm of ML. For example, proposing a learning algorithm that refines and adapts the proposed affordance extraction method during the learning process for a text-game RL agent?

**Strengths And Weaknesses:**

Strengths:
- Some of the error analysis and discussion of mistakes is interesting.
- The problem of automatically identifying affordances, and the motivation of discovering affordances in real-world action settings, is compelling.

Weaknesses:
- The contribution is not compelling. The paper is describing a simple application of ConceptNet to extract items in a text description of a scene (discovered via what is essentially teacher forcing through the game path), then identify tuples in ConceptNet relevant to those items. It is unclear whether the paper has relevance to TMLR, or whether there are any impacts beyond text-games.
- Even within the realm of text-games as an application, it's unclear how the proposed method would have impact. There is no evaluation of how the proposed method helps in the long-term goal of training agents that complete text-based games. What is the end task performance of an agent in these games given the proposed method?
- The precision metric is not well-motivated. If the goal is to recover additional feasible commands not explicitly specified by the game, it would be much more informative to evaluate recall of the original ACs rather than precision of the generated commands.
- It would be interesting to see what proportion of generated commands are actually useful towards the game's goal. Another question is: why were the original ACs chosen? Is it because they are feasible and helpful examples, or are they truly a subset of the potential reasonable actions to take?
- The human evaluation only covers a small number of generated commands, especially for TextWorld.
- Why are there remaining issues of generating "items not physically relevant/present in the current situation"? Are most of these problems happening because of other examples in the paper (e.g., items mentioned in a dream or memory), or for other reasons?
- Some missing details; see below

Missing details:
- What text corpus was used to extract prepositions, as mentioned in 4.2?

Smaller comments:
- Style violations; vertical space missing between paragraphs
- Citation for NELL is wrong. The paper described was published in 2015, not 2018, and there are older NELL papers as well (e.g. Carlson et al. 2010).
- Spelling mistakes (e.g., "proposition" instead of "preposition" in section 4.2).
- Use of "AC" as an acronym throughout is relatively confusing.

---

> ### Author Response · Authors · 2022-10-28
> **Reply & Withdrawal**
>
> Dear Reviewer,
>
> thank you very much for your helpful and concise feedback!
>
> We agree with you, that the connection to ML (and thus TMLR) is not as strong, as it probably should be. We were hoping to motivate the relevance of Affordance Extraction for ML applications (especially problem solving agents, which usually rely on an already provided action space), but reading the Reviews we understand that our work is located more in the fields of computer linguistic or KB research.
>
> We therefore decided to withdraw our submission and instead rework our paper according to your comments and proposals and look for submission in a thematically more suited field.
>
> Thank you again and best regards!

---

### Review · Reviewer_BLY6 · 2022-10-21

**Summary Of Contributions:**

The paper explores the use of large existing sources of data in the form of knowledge graphs (specifically ConceptNet) to generate commands in text games --- a relatively well known framework for testing multi-step language reasoning in grounded scenarios.

**Requested Changes:**

My requested changes will key off of the weaknesses mentioned above. Without such changes, the paper is not suitable for acceptance:
- Rewriting especially the related work and algorithm section, especially situating against prior work mentioned and how this work differs (if at all).
- Evaluating by using the action generation system introduced here in conjunction with a popular existing RL agent such as DRRN https://arxiv.org/abs/1511.04636 and comparing to existing RL agents (i.e. actually executing actions within the environment) or at the very least evaluating on a larger offline dataset such as TextWorld Express or JerichoWorld would help a reader analyze the relative benefits of this method.

**Strengths And Weaknesses:**

Strengths:
- The paper is well written overall and provides a good overview of the field for those less familiar with it
- The connection to cognitive science literature on affordances is made clear, providing the paper with a solid underlying motivation
- Evaluation on both Jericho and TextWorld (two of the primary text game benchmarks) makes it easier to see which domains this method works well in

Weaknesses:
- The paper is not situated with respect to former work on using knowledge bases for action generation in text games. The paper in its current form does not distinguish itself from these works and so it is unclear to a reader what exactly the novel contributions are here.
   - https://arxiv.org/abs/2005.00811 and https://www.aaai.org/AAAI21Papers/AAAI-7812.MurugesanK.pdf use knowledge from Concept to explicitly generate commands and run them in TextWorld, following https://arxiv.org/abs/1908.06556 who use ConceptNet as a source of commonsense information in text games
   - https://arxiv.org/abs/2010.02903 and https://arxiv.org/abs/2012.02757 look at using LLMs as a source of affordance information when generating text commands
- The evaluation setup is based on measuring the percentage of correctly generated actions using the walkthrough as opposed to executing actions within the environment as all online RL algorithm works in this area have done or using an offline pre-crawled dataset (of many states, not just those on the walkthrough - e.g. for TextWorld https://arxiv.org/abs/2208.01174 and for Jericho, the JerichoWorld dataset https://openreview.net/forum?id=7FHnnENUG0).
   - Using just the walkthroughs offers a very minimal setup for evaluating these games (a handful of a few hundred state/action pairs across the games where all previous works effectively evaluate on tens of thousands or more such state/action pairs). The low performance on the walkthroughs (even lower than the current state of the art performance for RL agents suggests much room for improvement in this method)

---

> ### Author Response · Authors · 2022-10-28
> **Reply & Withdrawal**
>
> Dear Reviewer,
>
> thank you very much for your concise and very helpful comments! We will use them as a guideline to improve our work.
>
> Still, after some discussion we decided to withdraw the submission and set another focus for a revised version. Running an RL agent on the created set of affordances is in principle possible, but would not yield any competitive/interesting results. We were hoping to point out the reasons for this (and encourage measures for improvement) and we will try to elaborate on this further in a revised version. We decided to follow the advice of another Reviewer and address another scientific community with stronger focus on computer linguistic problems, as we agree with you that the link to direct ML applications is probably a bit weak.
>
> For this we will also rework the literature section and i want to thank you especially for bringing the work of Murugesan et al. to our attention. Still we should remark, that while both paper follow a similar premise (utilizing ConceptNet for text-based environments), Murugesan et al. in my opinion address planing problems, not creation of an action space/affordance extraction. If i understood them correctly, they took already existing Admissible Commands provided by the TextWorld environment and used ConceptNet information to rate them. In contrast, our contribution does not address the process of planning, but the creation of an valid actionspace in the first place. We will try to emphasize this point in our new submission.
>
> Your comments helped us a lot to reflect on work, thank you again and best regards!

---

### Review · Reviewer_ho5b · 2022-10-22

**Summary Of Contributions:**

The manuscript attempts to solve the problem of affordances extraction in
text-based games by employing external, graph-based knowledge bases, presenting
an algorithm for matching potential knowledge tuples to the state-action space
of such games.



**Broader Impact Concerns:**

Generally this work doesn&rsquo;t seem to strictly require a broader impact statement,
although the authors might wish to consider addressing and/or discussing
possible biases embedded in their work due to the usage of external knowledge
bases.



**Requested Changes:**

Here's some broad feedback section by section that I hope helps setting the context for the overall review.

### Section 1

> Recently, the domain of&#x2026;

(nit) Not so recently, video games have pretty much been an AI benchmarking tool since
their inception.

Also, strictly speaking Go is not a video game per se.

1.  Typos

    1.  indespensible
    2.  up to our knowledge (not sure what this means? &ldquo;To the best of our knowledge&rdquo;?)
    3.  a major aspect for explaining (?)

### Section 2

> In practical applications this process is sometimes shortened or even omitted&#x2026;

I&rsquo;m not sure I&rsquo;m totally happy with the assumptions made in this paragraph. I
agree that affordances extractions is in principle a useful problem to solve,
but in principle there&rsquo;s no reason it needs to be a problem that is solved
explicitly. Most real life tasks and environments are likely to produce
affordances  that follow well known distributions (often, Zipf&rsquo;s law &#x2013; think
e.g. the autoregressive task in language modeling) that allow agents to
bootstrap to make a lot of trivial shortcuts. It is a relatively strong
proposition to claim that in RL unless one does explicit affordance extraction
one needs to rely on given prior information; it might be true, but I would try
to back the argument with some references.

> It should be noted that&#x2026;

This is also not a sound interpretation of much RL literature. Classically, an
MDP defines the task, no more (and no less!) than that. There&rsquo;s plenty of
literature that attempts to tackle cases where the action space is large (and
thus where exploration / action selection is a big problem simply due to scale).
See e.g.  <https://arxiv.org/abs/1906.12266> and literature therein.

I would suggest reworking this section to be a tad more fair to previous work,
or alternatively a stronger critique.

1.  Typos

    1.  as inGrabner&#x2026; (sic.)


### Section 3

> IF games offer a decreased complexity level due to a reduced amount of information&#x2026;

I&rsquo;m not sure I get what the manuscript is trying to say here. If it is trying to
define and compare &ldquo;complexity&rdquo; of tasks wrt. environmental dynamics, a good
option would be to compare (possibly abstracted) state spaces.

Similarly, it would be helpful to qualitatively compare Jericho vs TextWorld in
such terms in Section 3.1.2.

1.  Typos

    1.  byZelinka&#x2026; (sic.)


### Section 4

> &#x2026;as a possible resource for affordance extraction, several criteria have to be fulfilled:[&#x2026;]

It&rsquo;s perhaps more accurate to say that these are assumptions required by the
presented algorithm, rather than being strict requirements for the broader task
(one could e.g. imagine some degree of partial observability on data
availability that would still provide enough data in the general case).

> ConceptNet (CN) has been selected&#x2026;

Might be helpful to define CN where ConceptNet is first introduced, as it took
me some time to scan for the introduction of the acronym.

> For the purpose of affordance extraction on a basic level [CN weights] are
> ineffective most of the time.

In the general case, you might have some scale / compute problem, so wouldn&rsquo;t
they be extremely useful then? They&rsquo;d certainly provide some kind of search
prior, surely?

> The correct proposition is determined via statistical means by counting&#x2026;

This seems quite important, as it&rsquo;ll directly drive downstream results for
non-trivial n-tuples. It would be good to understand the algorithm used here,
and possible / seen failure cases;  this is important even if multi-object
commands are rare, because in the limit these might make for (potentially)
unfairly negative results.

Furthermore, I think it might be important to clarify that most affordances
tackled in this problem are fairly straightforward relationships earlier in the
manuscript, and that it might a potential downside of the approach /
methodology.

> The algorithm is written in Python 3.6.

Is this relevant information? Is there anything about this algorithm that
couldn&rsquo;t be written in other languages (or python versions!)?

> Figure 1

The manuscript might be better off using a textual / pseudocode / abstract
python representation for this algorithm (especially since details such as
logging seem to be relevant).

As a general comment, it is perfectly reasonable, and often &#x2013; also in this case
&#x2013; a good idea to focus on framework-level details rather than implementation
ones when describing an algorithm. For instance, the reader probably doesn&rsquo;t
really care <span class="underline">at this point of the manuscript</span> the **exact** data structure used to pass
information between functional steps.

> Textual description extraction then has to be adjusted accordingly.

This part is unclear and needs to be clarified. What does it mean to say that
&ldquo;every text source is valid&rdquo;? What problems does it create for the algorithmic
step?

> This is done when playing in TextWorld&#x2026;

It would be great to clarify whether the PoS-Tagging approach does indeed work
on TextWorld, or whether there are significant limitations to it for this
benchmark (and thus why the word list method was employed, which is far less
general).

> these disadvantages are accepted to retain a broader field of application.

What does this mean? An architectural / design problem of the algorithm doesn&rsquo;t
make said algorithm &ldquo;broadly applicable&rdquo;. If anything, it reduces its
*straightforward* applicability&#x2026;

> proceeding to the next scenario

This seems an important &ldquo;exploration&rdquo; step of the algorithm. How hard was it to
implement for the presented benchmarks, and what is the general complexity of
the problem?

To me it seems to be essentially a similar meta-task to the actual problem being
solved in the paper, and if so I wonder why one wouldn&rsquo;t be able to use the
affordance extractor in a similar scope and manner.

1.  Typos

    1.  may provide useful [information?]
    2.  It should be mentioned, that -> It should be mentioned that
    3.  recieve
    4.  is therefore able (not quite the right formulation here, semantics wise)

### Section 5

> Walkthrough option&#x2026;

See critique in earlier section.

> parser related constraints

To understand the evaluation choices, it is essential to expand on these and
defining what these constraints are. The following paragraphs essentially only
talks about the wordlist issue.

> as well as actions which have not been taken into account&#x2026;

Aren&rsquo;t these exactly the definition of affordances?

> A volunteer (one of the authors)

Considering the human baseline / evaluation seems to be extremely important for
the manuscript, it is important that the evaluation be unbiased and as
representative of the underlying problem as possible (especially since the
authors make a point about subjectivity being a potential issue). At such, I
would strongly suggest to employ multiple unrelated people (through e.g.
Mechanical Turk) rather than one of the authors.

> The next evaluation step addresses&#x2026;

This entire section is really unclear. The manuscript would be significantly
improved by clearly delineating and separating subsections by &ldquo;tasks&rdquo; and
algorithms used. It is really hard as it is to figure out what worked, what
didn&rsquo;t, and why. Section 5.4 is comparably easier to understand (but still needs
work).

**Strengths And Weaknesses:**

The manuscript attempts to solve an extremely interesting and relevant problem
for the RL / decision making community. It is generally hard to deal with large
action spaces, and language provides enough structure such that it might
be possible to strongly reduce a language-based action space.

I found the literature review and background sections to be decent and useful to
get perspective on the problem being tackled.

However, broadly speaking the manuscript needs significant work in multiple
areas:

1.  There&rsquo;s nothing that is significantly novel about utilising KBs to guide
    agents through structured (language) environments (see e.g.
    <https://arxiv.org/abs/2007.09185>).  Framing the problem as finding
    affordances might be, but the results are not strong enough to significantly
    make a dent towards forming a useful solution.
2.  The results seem to be largely negative (which is fine!), and whilst the
    authors have indeed put above-average effort in discussing tradeoffs and
    possible solutions, I don&rsquo;t believe the overall methodology goes past the
    community threshold. If this were a paper primarily focused on understanding
    why affordance extraction is hard in the proposed benchmarks, I would
    probably come to a different conclusion, but fundamentally it is a method
    paper (as it is presented).
3.  The evaluation, which effectively is what might be the most useful and
    impactful contribution of this work, is extremely untidy, and has significant
    design issues.

Finally, as it is I don&rsquo;t believe TMLR is  a great venue for this work, and the
algorithmic / KB communities in CL conferences might be a better target.
However, if the authors still wish to target ML folks, I would suggest
attempting to use and evaluate these affordances in an env / agent evaluation
setting, or to focus the narrative on improving [our understanding of] the
presented benchmarks.

---

> ### Author Response · Authors · 2022-10-28
> **Reply & Withdrawal**
>
> Dear Reviewer,
>
> thank you very much for your detailed, competent and very helpful comments! We will try to address as many points as possible/feasible for an revised version.
>
> Still, after a brief discussion we decided to withdraw our submission and follow your advice to target another scientific audience. We were hoping to motivate the relevance of Affordance Extraction for ML applications, but you are certainly right, that the connection to explicit ML applications is rather weak at this point. Running an ML agent on the created set of affordances would not yield any good results at this point (for reasons which were partly discussed in the paper but surely could have been made clearer). We will change the focus of the revised paper accordingly.
>
> Thank you again for your feedback and best regards!

---

### Note · Authors · 2022-10-30

**Comment:**

Following the usefull feedback from the Reviewers we decided to improve the study based on the given suggestions and submit to a venue which is more suitable for the topic of Affordance Extraction.

Thank you very much and best regards

**Withdrawal Confirmation:**

I have read and agree with the venue's withdrawal policy on behalf of myself and my co-authors.